# Polyploids broadly generate novel haplotypes from trans-specific variation in *Arabidopsis arenosa* and *Arabidopsis lyrata*

Magdalena Bohutínská[1,2,3]*, Eliška Petříková[1], Tom R. Booker[4], Cristina Vives Cobo[1], Jakub Vlček[1,5], Gabriela Šrámková[1], Alžběta Poupětová[1], Jakub Hojka[1,2], Karol Marhold[1], Levi Yant[1,6◉], Filip Kolář[1,2◉], Roswitha Schmickl[1,2◉]*

1 Department of Botany, Faculty of Science, Charles University, Prague, Czech Republic, 2 Institute of Botany, Czech Academy of Sciences, Průhonice, Czech Republic, 3 Institute of Ecology and Evolution, University of Bern, Bern, Switzerland, 4 Department of Forest and Conservation Sciences, University of British Columbia, Vancouver, British Columbia, Canada, 5 Biology Centre, Czech Academy of Sciences, České Budějovice, Czech Republic, 6 Department of Life Sciences, University of Nottingham, Nottingham, United Kingdom

◉ These authors contributed equally to this work.
* magdalena.bohutinska@natur.cuni.cz (MB), roswitha.schmickl@natur.cuni.cz (RS)

**Data Availability Statement:** Sequence data that support the findings of this study are deposited in the NCBI (https://www.ncbi.nlm.nih.gov/bioproject/) under BioProjects PRJNA284572, PRJNA309929,

## Abstract

Polyploidy, the result of whole genome duplication (WGD), is widespread across the tree of life and is often associated with speciation and adaptability. It is thought that adaptation in autopolyploids (within-species polyploids) may be facilitated by increased access to genetic variation. This variation may be sourced from gene flow with sister diploids and new access to other tetraploid lineages, as well as from increased mutational targets provided by doubled DNA content. Here, we deconstruct in detail the origins of haplotypes displaying the strongest selection signals in established, successful autopolyploids, *Arabidopsis lyrata* and *Arabidopsis arenosa*. We see strong signatures of selection in 17 genes implied in meiosis, cell cycle, and transcription across all four autotetraploid lineages present in our expanded sampling of 983 sequenced genomes. Most prominent in our results is the finding that the tetraploid-characteristic haplotypes with the most robust signals of selection were completely absent in all diploid sisters. In contrast, the fine-scaled variant 'mosaics' in the tetraploids originated from highly diverse evolutionary sources. These include widespread novel reassortments of trans-specific polymorphism from diploids, new mutations, and tetraploid-specific inter-species hybridization–a pattern that is in line with the broad-scale acquisition and reshuffling of potentially adaptive variation in tetraploids.

## Author summary

Polyploidy, the result of whole genome duplication, is associated with speciation and adaptation. To fuel their often remarkable adaptations, polyploids may access and maintain adaptive alleles more readily than diploids. Here, we identify repeated signals of selection on genes that are thought to mediate adaptation to whole genome duplication in two

PRJNA357693, PRJNA357372, PRJNA459481, PRJNA493227, PRJEB34247 (ENA), PRJNA506705, PRJNA484107, PRJNA592307, PRJNA667586, PRJNA929698. See S1 Data for individual codes. ScanTools_ProtEvol pipeline: github.com/mbohutinska/ScanTools_ProtEvol ABBA-BABA pipeline: github.com/simonhmartin/tutorials/tree/master/ABBA_BABA_whole_genome PicMin: github.com/TBooker/PicMin Allele frequencies of haplotype blocks: github.com/mbohutinska/repeatedWGD, section 'Haplotype AF'.

**Funding:** This work was supported by the Czech Science Foundation (project 20-22783S to F.K., project 19-06632S to K.M.), Leverhulme Trust award (no. RPG-2020-367 to L.Y.), PRIMUS Research Programme of Charles University (PRIMUS/17/SCI/23 to R.S.), European Union's research and innovation programme under the Marie Skłodowska-Curie (project 101062703 to M. B.), European Research Council (project 850852 DOUBLEADAPT to F.K.), Charles University Grant Agency (no. 219223 to A.P.), and long-term research development project no. RVO 67985939 of the Czech Academy of Sciences. The funders had no role in study design, data collection and analysis, decision to publish, or preparation of the manuscript.

**Competing interests:** The authors have declared that no competing interests exist.

*Arabidopsis* species. We found that the tetraploid-characteristic haplotypes, found in genes exhibiting the most robust signals of selection, were never present in their diploid relatives. Instead, these haplotypes were made of novel 'mosaics' forged from multiple allelic sources. We hypothesize that this increased variation forms the genic basis of the potentially eased adaptation of polyploids.

## Introduction

Whole genome duplication (WGD) is widespread across eukaryotes, especially in plants. It comes with significant costs, such as meiotic instability and cell cycle changes, both of which require adaptation to WGD [1,2]. Over the last decade, a body of work has accrued describing means by which within-species WGD lineages (taxonomic autopolyploids) or WGD lineages with polysomic inheritance (random chromosome pairing; genetic autopolyploids [3,4]) overcome these challenges. The most obvious of these appears to be a stabilization of cell division via selection acting at meiotic and mitotic genes [5–8]. Additionally, cyclin genes have been observed to mediate tolerance to tetraploidization in polyploid tumors and in proliferating *Arabidopsis* tissues [9–12]. While certain genes involved in adaptation to WGD have been identified and their functions verified [13–15], a broader evolutionary context remains elusive. Questions persist regarding the origins of this adaptive genetic variation and the mechanisms by which these variants assemble into positively selected haplotypes in nascent polyploids.

In diploids, various sources contribute to the pool of adaptive alleles, including gene flow/introgression, ancestral standing variation, and de novo mutations (Fig 1A). Recent work indicates the potential for all these sources to become greater following WGD (Fig 1A). First, polyploid lineages can benefit from the weakening of hybridization barriers [16–18], leading to increased allelic exchange through introgression [19,20]. Polyploids can also gain additional diversity via introgression from sister diploids, whereas introgression in the opposite direction is much less likely due to the nature of the diploid-tetraploid barrier [21–24]. Second, polyploids maintain genetic diversity more efficiently due to increased masking of recessive alleles [25,26], thus keeping a larger pool of standing variation [27]. Finally, polyploids experience elevated mutational input due to the doubled chromosome numbers, leading to a higher number of novel mutations per generation [28]. Consequently, autopolyploids are hypothesized to generate and maintain a greater degree of genetic variability than diploids.

Traditionally, beneficial alleles have been envisioned as originating from a single source (Fig 1B), either from introgression, standing variation, or de novo mutations [29–33]. However, recent evidence suggests that alleles can also gradually accrue to form finely tuned haplotypes [34–36]. This implies that adaptive haplotypes may accumulate from multiple sources, rather than just one (Fig 1B). Here, we ask if polyploids use their expanded allelic sources to construct novel fine-scaled 'mosaic' haplotypes of diverse origins, what those exact origins are, and if the genes involved in adaptation to WGD form an exception compared to the genomic background.

The naturally ploidy-variable (diploid and autotetraploid) species *Arabidopsis arenosa* and *Arabidopsis lyrata* have emerged as powerful models to understand adaptation to WGD [37]. Recent research in these species identified shared candidate alleles mediating adaptation to WGD, which were shared between *A. arenosa* and *A. lyrata* by a process of tetraploid-specific adaptive introgression [5,7,20,35]. There have been indications that tetraploidy-related candidate adaptive alleles might have originated from specific source populations, or that recombination may be involved in the construction of adaptive alleles [20,35]. However, these were

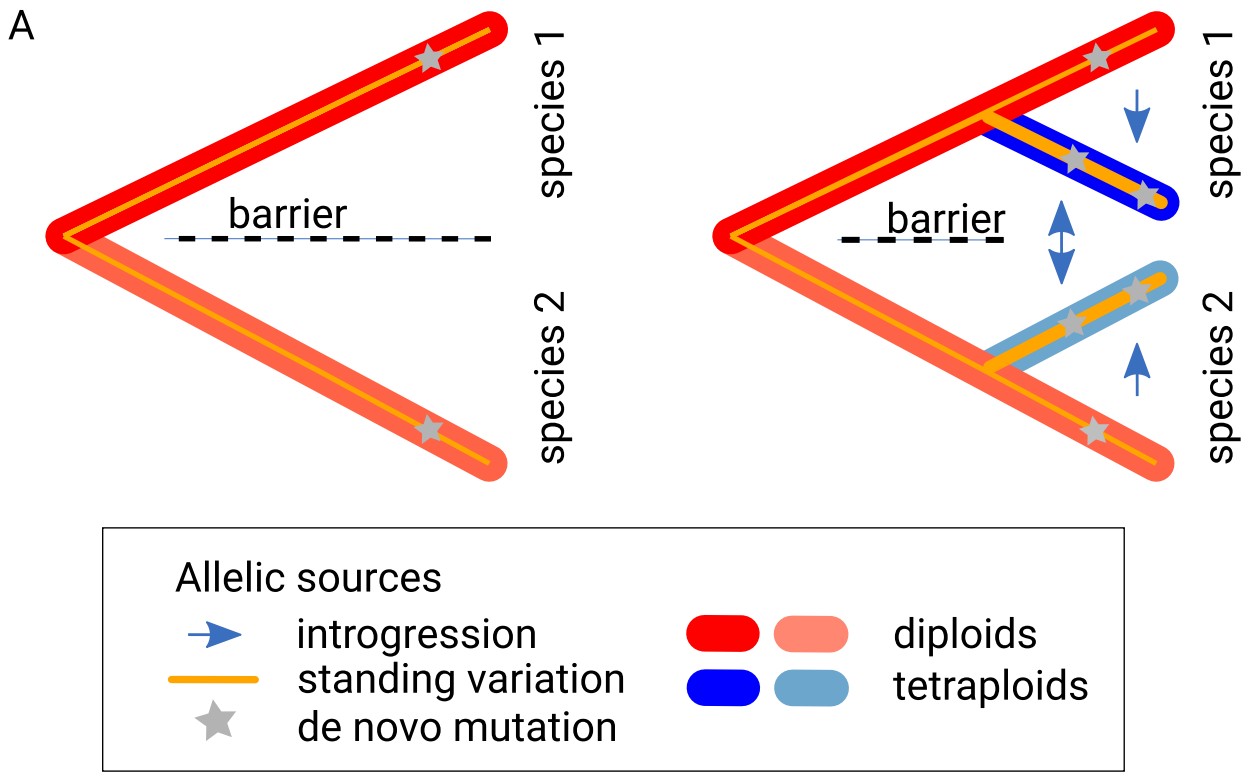

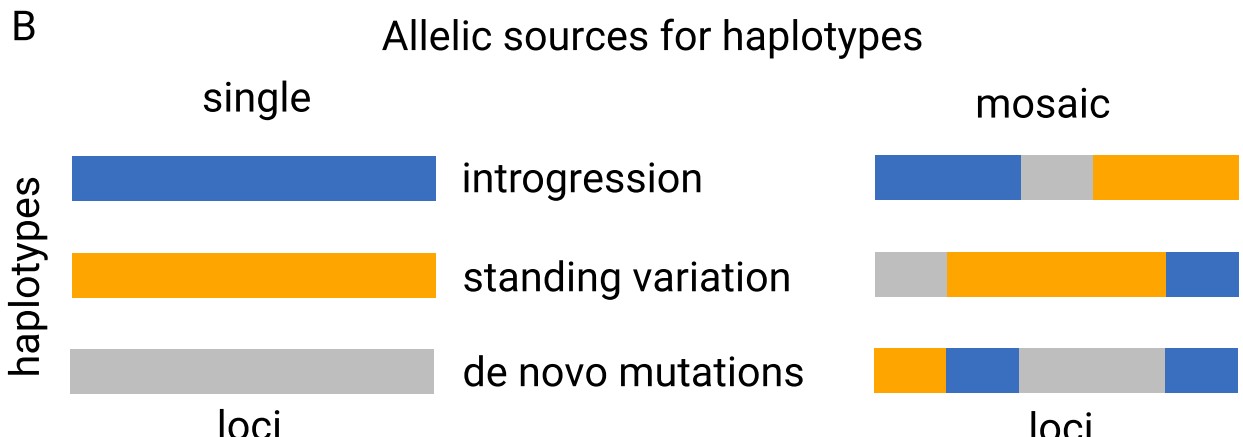

**Fig 1. Hypotheses about allelic sources in a diploid-autotetraploid system.** A: As compared to diploids, autotetraploid lineages may acquire alleles via increased introgression potential, higher level of retainment of ancestral standing variation, and increased population-scaled mutational input. Shown are two diploid species (red), each of which gave rise to an autotetraploid lineage (blue). B: Higher variability in the possible sources of potentially beneficial alleles suggests that positively selected haplotypes in tetraploids might be more likely to form from a mosaic of sources (right), in contrast to the traditionally assumed homogeneous (single-source) scenario (left).

based on small sample sizes, and the studies did not consider a role for trans-specific polymorphism (here represented by ancestral alleles shared among different *Arabidopsis* species).

Here, we take advantage of an exhaustive dataset of 983 sequenced individuals encompassing all known lineages of European *Arabidopsis* autotetraploids, backed by genome-wide

diversity of all diploid outcrossing *Arabidopsis* species. Apart from the tetraploid *A. lyrata* lineage from the eastern Austrian Forealps ('Austria' hereafter; [20,38]), we newly genomically characterized two additional *A. lyrata* tetraploid lineages from Central Europe: south-eastern Czechia ('Czechia' hereafter) and Harz in Germany ('Germany' hereafter). We performed a joint analysis of these three *A. lyrata* tetraploid lineages and tetraploid *A. arenosa*, which derived from a single WGD event [26,39]. We used this sampling of four tetraploid lineages, established which genes and processes are robustly under selection in all of them, and elucidated the sources of candidate adaptive alleles to WGD. Using this knowledge, we asked to which extent tetraploid-specific haplotype blocks (referred to as 'haplotypes' further) were formed and what their evolutionary sources were.

## Results

### Introgression between all four autotetraploid *Arabidopsis* lineages in Central Europe

We found that *A. lyrata* autotetraploids occur at geographically distinct locations throughout Central Europe (Fig 2A and S1 and S2 Data). To obtain comparable and representative samples of *A. lyrata* and *A. arenosa* populations for diploid-tetraploid selection scans, we first analyzed 154 individuals from 17 proximal diploid and tetraploid populations (Fig 2A). We assessed population structure and potential admixture using bootstrapped allele covariance trees (TreeMix [40]; Figs 2B and 2C), principal component analysis (S1 Fig), neighbor-joining networks (SplitsTree [41]; S1 Fig), and Bayesian clustering (fastSTRUCTURE [42]; S1 Fig). In *A. arenosa*, we identified a single Central European tetraploid lineage (Fig 2B) in line with previous range-wide studies [26,39]. Notably, *A. lyrata* tetraploids consisted of three differentiated lineages in Austria, Czechia, and Germany (Fig 2C). Both species and cytotypes maintained high genetic diversity (*A. arenosa*: mean nucleotide diversity over four-fold degenerate sites (4d-π) = 0.026/0.024 for diploids/tetraploids; *A. lyrata*: mean 4d-π = 0.012/0.016 for diploids/tetraploids; S1 Table). Although Central European *A. lyrata* has a patchier distribution and showed lower nucleotide diversity than *A. arenosa*, both species and cytotypes exhibit neutral Tajima's D, indicating their populations are very close to neutrality (mean Tajima's D in *A. arenosa* = 0.01/0.20 for diploids/tetraploids; mean Tajima's D in *A. lyrata* = 0.27/0.23 for diploids/tetraploids; S1 Table).

Next, we examined introgression between the two species using D-statistics (ABBA-BABA statistics [43]). While insignificant among diploids, we identified evidence of introgression (D between 0.106 to 0.108; Fig 2D) between tetraploids of *A. arenosa* and each of the three tetraploid *A. lyrata* lineages.

### Genomic signatures of repeated differentiation associated with WGD

To infer candidate genes associated with signatures of selection in diploid-tetraploid contrasts, we compared each of the four tetraploid *A. lyrata*/*A. arenosa* lineages to their geographically most proximal diploids (Figs 2A–2C). Using PicMin [44], which measures repeatedly selected regions in genomic windows, we identified 54 repeatedly differentiated windows (PicMin FDR-corrected *q*-value < 0.01; S3 Data), overlapping 14 unlinked candidate genes (Fig 2E). To capture narrower peaks of differentiation, we also conducted a search for genes featuring highly diploid/tetraploid differentiated SNPs present in multiple lineages (see Methods). Such a 'candidate SNP' approach can identify genes undergoing positive selection in tetraploids on a subset of SNPs while the evolution of the rest of the gene sequence is constrained. This analysis, with 1% outlier Fst cutoff, identified all 14 PicMin candidate genes and three additional

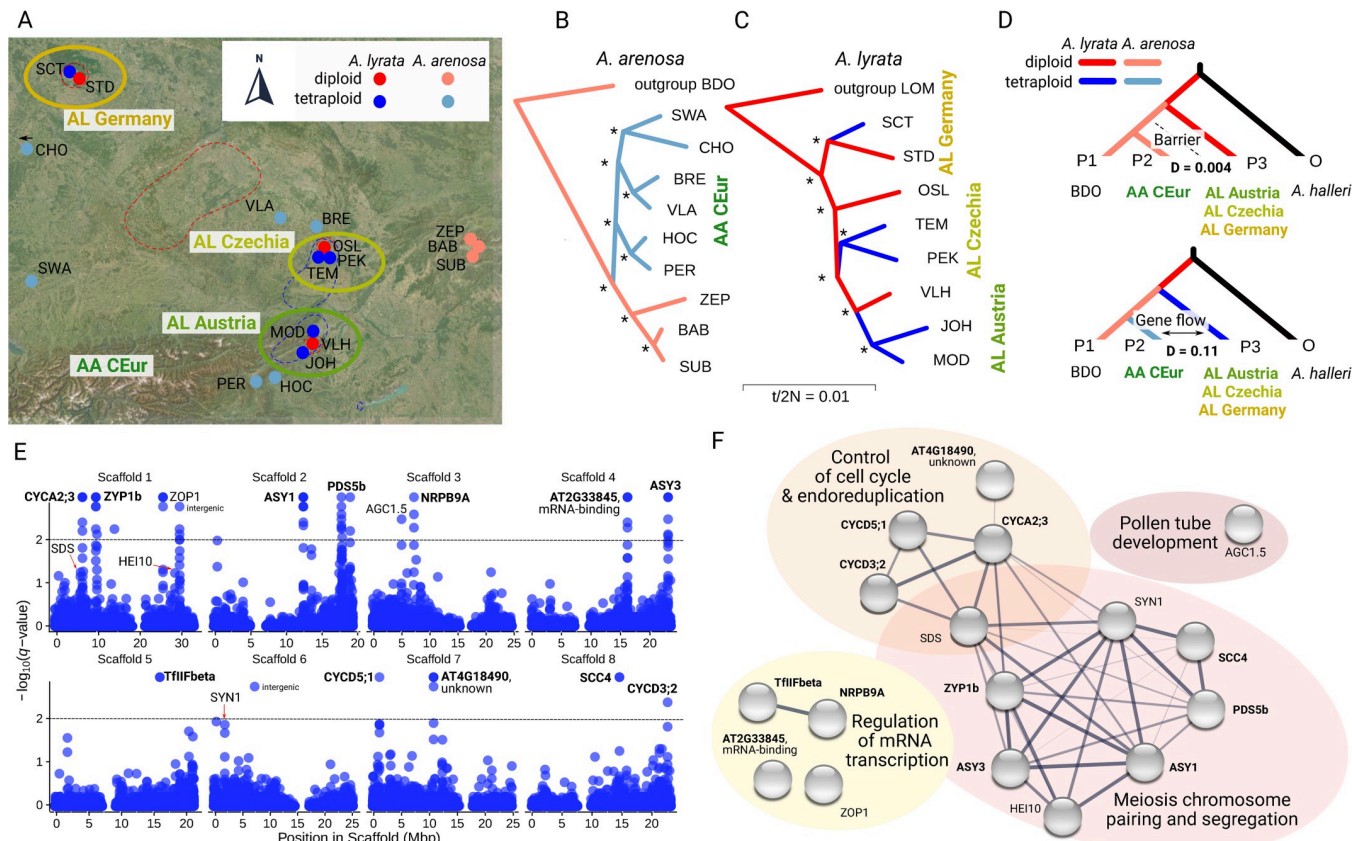

**Fig 2. Evolutionary relationships and genomic signatures of selection in tetraploid populations of *A. lyrata* and *A. arenosa*.** A: Locations of the focal 11 *A. lyrata* populations (AL Austria, AL Czechia, and AL Germany) and six *A. arenosa* populations (AA CEur) in Central Europe. Red and blue dashed lines show the ranges of diploid and tetraploid *A. lyrata*, respectively; tetraploids of *A. arenosa* occur throughout the entire area. For the map, the base layer 'World Imagery' was taken from https://www.usgs.gov/products/, which is granted to be published under CC BY 4.0 license. B, C: Phylogenetic relationships among the focal populations of *A. arenosa* (B) and *A. lyrata* (C) inferred by TreeMix analysis assuming no migration. Asterisks show bootstrap support = 100%. D: Introgression among tetraploid but not diploid populations of both *Arabidopsis* species. ABBA-BABA statistics demonstrating excess allele sharing between tetraploids (bottom tree), but not diploids (top tree) of *A. arenosa* and each of the three *A. lyrata* tetraploid lineages. P1 is BDO, the earliest diverging and spatially isolated diploid population of *A. arenosa*. E: A set of 14 unlinked genes showing significant evidence (p < 0.01) of positive selection (positively selected genes, PSGs) in the four tetraploid lineages identified using PicMin. Additional three genes, *HEI10*, *SDS*, and *SYN1*, were identified using a screen for candidate SNPs. F: Functional characterization of the 17 PSGs by STRING analysis. The network shows predicted protein-protein interactions among the PSGs. The width of each line corresponds to the confidence of the interaction prediction. PSGs were annotated into four processes, each represented by a bubble of different color. The 12 PSGs with names written in bold had enough candidate SNPs for the reconstruction of tetraploid haplotypes (see the main text).

candidates (*HEI10*, *SDS*, and *SYN1*), each with multiple outlier SNPs differentiated across all four tetraploid lineages. In total, we recognized 17 candidate positively selected genes (PSGs) showing robust, repeated selection signals in tetraploid lineages (S2 Table). Notably, these loci do not cluster in regions characterized by extreme recombination rates (S2 Fig), indicating no bias associated with the recombination landscape [45].

To address in which molecular processes the 17 WGD-associated PSGs are involved, we predicted protein-protein interactions among them using STRING [46]. We retrieved two interconnected clusters: control of cell cycle and endoreduplication, represented by the genes *CYCA2;3*, *CYCD3;2*, *CYCD5;1*, and chromosome pairing and segregation during meiosis, with the genes *ASY1*, *ASY3*, *HEI10*, *PDS5b*, *SCC4*, *SDS*, *SYN1*, *ZYP1b* (p < 0.001, Fisher's exact test; Fig 2F and S3 Table). An additional six of the 17 PSGs were not connected in this interaction network. Four of these six genes were found to be related to mRNA transcription

(*AT2G33845*, *NRPB9A*, *TfIIFbeta*, *ZOP1*; Fig 2F). *AT4G18490* encodes an unknown protein that is strongly expressed in young *A. thaliana* flower buds and involved in cell division [47]. *AGC1.5* was not connected to any of these processes. It has recently been shown to be involved in maintaining functional pollen tubes in tetraploid *A. arenosa* [15]. We provide further functional interpretations of the candidate PSGs in S1 Text. The candidate PSGs varied in their proportion of differentiated cis-regulatory, nonsynonymous, and synonymous SNPs (S3 Fig and S1 Text).

Some phenotypic shifts in polyploids have been associated with the above identified molecular processes in *Arabidopsis*. Specifically, cytological studies reported stable meiotic chromosome segregation in established tetraploids of *A. arenosa* and *A. lyrata* compared to neo-tetraploids of *A. arenosa*, which suggests a compensatory shift in the meiotic stability phenotype [5,13,14,20,35]. Here, we investigated whether endoreduplication follows a similar pattern of compensatory evolution after WGD, in line with the suggestion that "whole-plant polyploidy may have a subtle inhibitory effect on the extent to which cells undergo endopolyploidy" [1]. Using flow cytometry, we found a substantial decrease in endoreduplication within established tetraploids compared to synthetic tetraploids, while accounting for the variation in diploids (S4 and S5 Figs and S2 Text). This finding suggests compensatory changes, possibly reflecting a high cost of additional rounds of genome duplication in polyploids and/or structural constraints on maximum cell size [48,49]. It should be highlighted, however, that causality between the observed phenotype and variation in the relevant PSGs needs further functional validation.

## Tetraploid haplotypes are assembled from diverse allelic sources

Next, we tested whether these positively selected regions in tetraploids aggregate as mosaics from disparate allelic sources (Fig 1B): To do this, we first constructed the haplotypes of the PSGs and identified the specific allelic sources for every candidate tetraploid SNP. Removing five PSGs with insufficient variation (less than five candidate SNPs), we analyzed 12 genes comprising in total 232 candidate SNPs (S4 Data). For each allele, we reconstructed diploid- or tetraploid-characteristic haplotype blocks using these candidate SNPs as markers following [7]. These haplotype blocks corresponded to physical haplotypes observed in PacBio HiFi reads from five diploid and five tetraploid individuals of *A. arenosa* (71% correspondence across 132 checked long reads, 95% correspondence if heterozygous sites were excluded; S5–S7 Data), and we term them 'haplotypes' hereafter for simplicity.

We identified a single major tetraploid-characteristic haplotype for each of the 12 candidate genes that was shared across both *A. lyrata* and *A. arenosa* tetraploids (present in 92% and 98% of populations, respectively, mean frequency = 0.62 across 61 tetraploid populations; Fig 3B and S4 Table). We further identified two major diploid haplotypes for each gene, one *A. lyrata*-characteristic and one *A. arenosa*-characteristic (Fig 3B; see Fig 3A for the locations of populations of the different *Arabidopsis* species and cytotypes used in this analysis). Other haplotypes were of minor frequency (Fig 3B). Importantly, none of the tetraploid haplotypes associated with the 12 PSGs were found in any diploid individuals, even though our sampling included all of the known potential source lineages for the tetraploids.

Despite the absence of each of the 12 tetraploid haplotypes (corresponding to the 12 PSGs) in diploids, certain tetraploid SNPs defining these haplotypes were detected across diploid populations, present in one or even across multiple species (Fig 4). We therefore investigated the possibility that the tetraploid haplotypes might be assembled from multiple evolutionary sources (introgression, standing variation, de novo mutations). To do this, we inferred the most likely source for each candidate SNP marking these tetraploid haplotypes. Considering

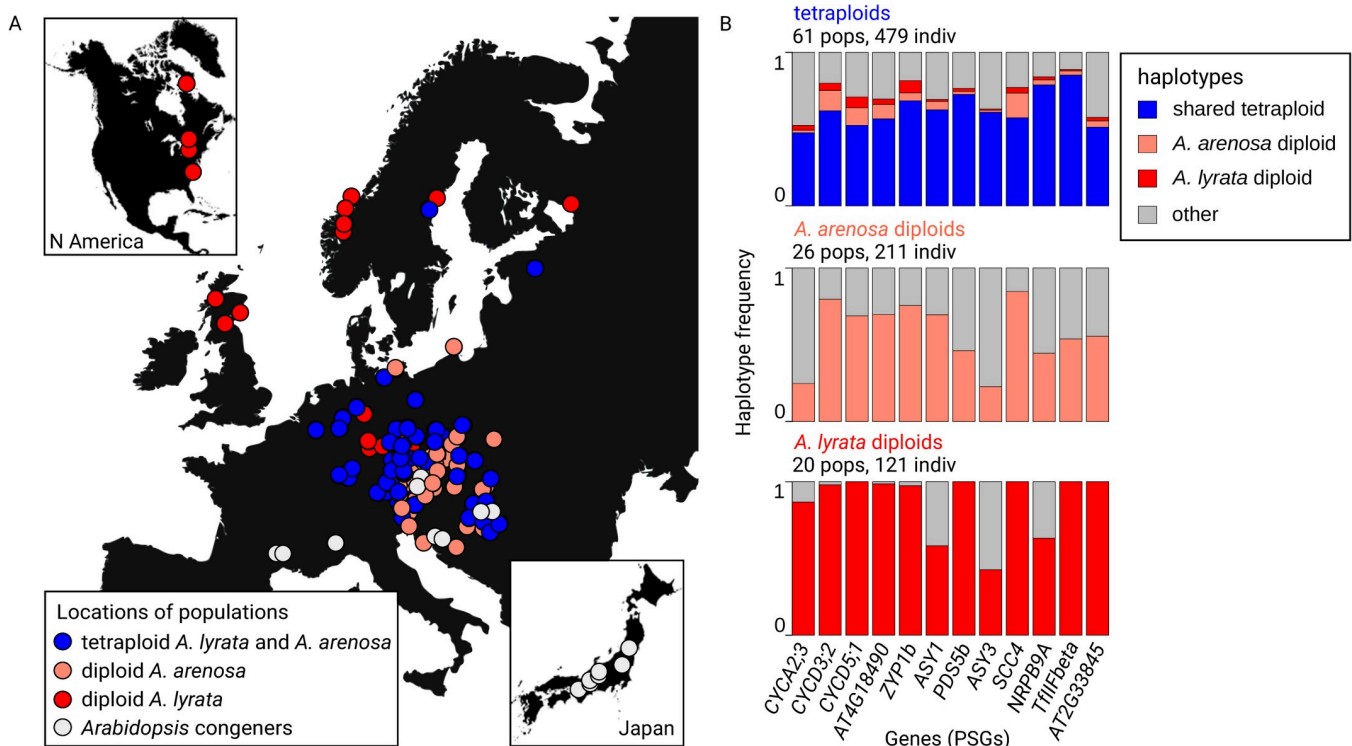

**Fig 3. Distribution of haplotypes of the 12 positively selected genes (PSGs) involved in adaptation to WGD.** A: Populations in this analysis (504 diploids and 479 tetraploids). *Arabidopsis* congeners are *A. halleri*, *A. croatica*, *A. cebennensis*, and *A. pedemontana*. The map was vectorized and modified according to the base layer 'Dark Gray canvas', https://www.usgs.gov/products/, which is granted to be published under CC BY 4.0 license. B: Frequency of shared tetraploid (blue), *A. arenosa* diploid (light red), and *A. lyrata* diploid (red) haplotypes in each of the 12 PSGs. All other haplotypes present (including possible recombinants of the above) are shown in grey. Note the absence of tetraploid haplotypes in diploids.

the extent of trans-specific polymorphism in outcrossing *Arabidopsis* [50], we worked with the full dataset of *A. arenosa*, *A. lyrata*, and all other diploid *Arabidopsis* outcrossers (*A. halleri*, *A. croatica*, *A. cebennensis*, and *A. pedemontana*) (S1 and S2 Data). Using the phylogenetic relationships among these species ([51], Fig 4A) and the SNP presence/absence data from all 504 diploid individuals, we identified the most likely source for each of the 232 candidate SNPs (Fig 4 and S8 Data). The different allelic sources forming tetraploid haplotypes varied genome-wide (when analyzing all 12 PSGs together), but also within individual tetraploid haplotypes (3–6 scenarios per haplotype, median = 4, out of 7; Figs 4 and S6). Overall, 65.5% of the SNPs forming tetraploid haplotypes were most parsimoniously inferred as trans-specific polymorphism, segregating in diploids of multiple *Arabidopsis* species (Fig 4A, scenarios 1–4). Note that this number is likely an underestimate as some alleles may have remained unsampled or have gone extinct in any diploid lineage. Only 6.9% of the candidate SNPs were inferred as arising from standing variation from a single diploid progenitor (*A. arenosa* or *A. lyrata*; Fig 4A, scenarios 5, 6), and the remaining 27.6% of SNPs possibly accumulated de novo in tetraploids or remained unsampled in our dataset (absent in any of the 504 diploid individuals; Fig 4A, scenario 7). Detailing trans-specific polymorphism, we further quantified that 35.8% of the candidate SNPs marking the tetraploid haplotypes were likely contributed from diploid *A. arenosa* (present in this species and possibly in diploids of other species, but not *A. lyrata*; Fig 4A, scenarios 2, 5), while 3.4% likely came from diploid *A. lyrata* (Fig 4A, scenarios 3, 6). Finally, in the process of tetraploid haplotype formation through these identified source scenarios, 71.1% of the candidate SNPs were indicative of interspecific introgression across the

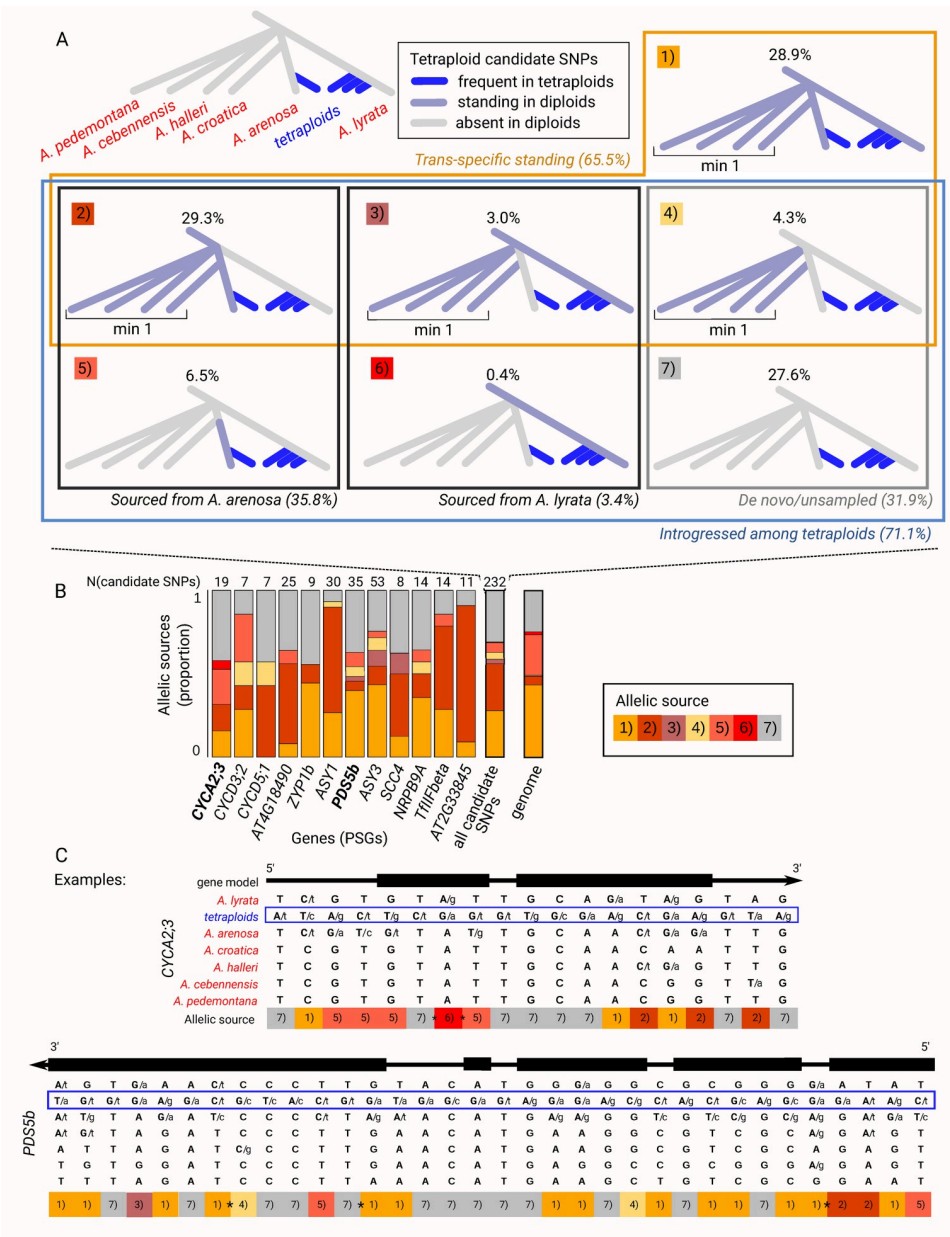

**Fig 4. Mosaic of allelic sources of tetraploid haplotypes corresponding to the 12 positively selected genes (PSGs) associated with adaptation to WGD.** A: The 232 candidate SNPs marking the 12 tetraploid haplotypes were categorized into one of the seven source scenarios based on their allele distribution in the 983 samples. The most parsimonious origin of each pattern is provided in italics. Four scenarios (outlined by orange frame) involve trans-specific standing variation shared among diploids of at least two *Arabidopsis* species ('standing in diploids'). Six scenarios (outlined by blue frame) require introgression between tetraploid lineages. Boxes 'min 1' show that the tetraploid standing variation is present in at least one outgroup species. Phylogenetic relationships according to [51], Fig 4A. B: Variable source scenarios for each of the 12 PSGs. Barplots show the proportion of candidate SNPs representing each of the seven source scenarios, numbers above bars indicate the number of candidate SNPs per each PSG. Two example PSGs detailed in panel C are highlighted in bold. C: Illustration of diploid and tetraploid haplotypes for two PSGs. Bold capital letters represent haplotype-defining alleles, and two letters at a position indicate the presence of an alternative minor allele. Coding sequences are highlighted as black boxes as part of the gene model above the haplotypes (only regions overlapping with candidate SNPs are shown). The bottom line 'allelic source' displays the SNP assignment to its source scenario as defined in panel A. Possible recombination breakpoints required to construct the haplotype from the different allelic sources are marked with asterisks. Top: example from the *CYCA2;3* endoreduplication gene, displaying 19 candidate SNPs spanning 5371 bp. Bottom: example from the *PDS5b* meiosis gene, depicting candidate SNPs spanning 8904 bp.

tetraploid lineages. This was inferred from their presence in tetraploids and in at most one of the diploid progenitors (either diploid *A. arenosa* or *A. lyrata*; note that this scenario is nested within others; Fig 4A, scenarios 2–7).

In summary, this analysis shows that tetraploid haplotypes represent a combination of reuse of trans-specific as well as species-specific SNPs, with additional input of de novo mutations after WGD (Figs 4B and 4C). The mosaic nature of allelic sources was further supported by the reticulation observed in the neighbor-joining networks of the 12 PSGs (S7 Fig). In these networks, tetraploids form a single lineage that connects through multiple splits to multiple diploid *Arabidopsis* lineages. This variability of allelic sources was also found for the genomic background (of 232 randomly sampled synonymous SNPs sampled 1000 times). Interestingly, however, the proportion of trans-specific polymorphism in the 232 candidate SNPs was higher compared to the genomic background average (65.5% compared to 50.0%, respectively, p-value = 0.001, 1000 permutations; Fig 4A and S5 Table). Similarly, the proportion of SNPs indicative of introgression across the tetraploid lineages was much higher for the candidate SNPs than for the average of the genomic SNP subset (71.1% compared to 56.4%, respectively, p-value = 0.002, 1000 permutations; Fig 4A and S5 Table). Finally, the proportion of SNPs exclusively present in tetraploids was slightly but nonsignificantly higher for the 232 candidate SNPs compared to the average of the genomic background (27.6% compared to 24.5%, respectively, p-value = 0.493, 1000 permutations; Fig 4A and S5 Table).

## Discussion

To exhaustively assess sources of adaptive variation in young autotetraploids of *A. arenosa* and *A. lyrata*, we sampled all regions possibly harboring the tetraploid cytotype, focusing on Central European *A. lyrata* [20,35,38,52] and all European diploid outcrossing *Arabidopsis* species. Notably, *A. lyrata* tetraploids consisted of three differentiated lineages in Austria, Czechia, and Germany that were genetically close to parapatric diploid populations of the same species. This suggests either independent formation and establishment of tetraploid populations in each region, or a single origin of autotetraploid *A. lyrata* followed by allopatry and rampant local introgression from parapatric diploid *A. lyrata* lineages. In *A. arenosa*, we identified a single Central European tetraploid lineage, consistent with previous studies [26,39]. Our finding of introgression between tetraploids of *A. arenosa* and each of the three tetraploid *A. lyrata* lineages is in line with previous experimental studies: diploids of both species exhibit strong postzygotic barriers, whereas polyploidy-mediated hybrid seed rescue (i.e. reestablishment of endosperm cellularization through polyploidy [17]) enables hybridization between tetraploid *A. arenosa* and *A. lyrata* [20,38]. Overall, we identified four tetraploid lineages in Central European *Arabidopsis*, consistent with two to four independent WGD events (one in each species and possibly up to two additional WGD events in *A. lyrata*).

We found 17 PSGs shared across the tetraploid lineages, mediating processes of homologous chromosome pairing during prophase I of meiosis, cell cycle timing and regulation of endoreduplication via different classes of cyclins, and mRNA transcription via RNA polymerase II. A similar set of WGD-associated candidate genes was found in [5] for *A. arenosa* and in [20] for the Austrian lineage of *A. lyrata*. The interacting meiosis and cell cycle regulation PSGs likely mediate beneficial phenotypic shifts resulting in the establishment of tetraploid lineages. Previous work reported relatively stable meiotic chromosome segregation in established tetraploids of *A. arenosa* compared to newly synthesized tetraploids of *A. arenosa* [5,13,14], indicating compensatory evolution following WGD. Here, we show signals of such compensatory evolution also in the case of endoreduplication, as established *A. arenosa* tetraploids exhibited a decrease in endoreduplication compared to newly synthesized tetraploids.

Additional mechanistic work is required to understand the functional and strategic basis of this shift. It may be a homeostatic response to maintain cell size, overall DNA content, or mitotic stability at high ploidies [48,49]. Recent studies have repeatedly demonstrated that certain genes involved in meiosis and cell cycle regulation are the core mediators of adaptation to WGD in *A. arenosa* and *A. lyrata* [5,20]. Yet, additional processes, like ion management [6,8] or pollen tube growth [15], have been found to play a role in adaptation to WGD across the family Brassicaceae, suggesting that alternative solutions to similar challenges may exist. Where the previous literature stops, however, is in detailed sourcing of the candidate adaptive alleles and their exact origins, which is here recognized as highly mosaic. Indeed, both theoretical and empirical evidence indicates that polyploids may have increased access to allelic variation than diploids [19,25,26,28]. In *Arabidopsis* tetraploids it is largely ancestral standing variation in the form of trans-specific polymorphism, segregating in diploids of multiple *Arabidopsis* species, that sources allelic variation in the PSGs. This supports the growing recognition of the role of trans-specific standing polymorphism in adaptation [53,54]. Introgression among tetraploids also substantially contributes to allelic variation in tetraploids, aligning with the observed signatures of gene flow among tetraploids of both species [20,38] and with the general recognition of introgression to generate potentially beneficial diversity within polyploid species complexes (reviewed in [16,18,55]). De novo mutations appear to have a subordinate role in tetraploid allelic variation, although they may nevertheless be crucial for rapid protein (co-)evolution during meiotic adaptation, as indicated in *A. arenosa* [7].

Overall, each haplotype corresponding to the PSGs comprised SNPs of varying ancestry, including trans-specific and species-specific polymorphisms from closer and more distant relatives, along with likely de novo mutations in the tetraploids. These 'mosaic' haplotypes were then likely shared via introgression among tetraploids and spread across Europe. Variable allelic sources were also found for the genomic background (i.e. subset of randomly selected synonymous SNPs), although trans-specific polymorphism, introgression, and de novo mutations in tetraploids were more frequent for the candidate SNPs. This finding is, however, biased by the fact that we sampled only candidate SNPs that repeatedly overlapped between *A. lyrata* and *A. arenosa*, as the PSGs constitute a subset of genes replicated across all tetraploid lineages. It is therefore expected that the proportion of introgression-indicative variants is higher in the PSGs compared to the genomic background. Most of the trans-specific polymorphism for both the candidate SNPs and the genomic background SNPs stem from *A. arenosa* and at least one outgroup species, and we hypothesize that this is due to the basal position of *A. arenosa* in a novel phylogeny of the genus *Arabidopsis* that we recently generated. This distinct phylogenetic placement of *A. arenosa* both enabled accumulation of private alleles and allele sharing with derived species. In addition, the high nucleotide diversity of diploid *A. arenosa* (on average twice of that in *A. lyrata*) allowed for retention of a large allelic pool to serve as raw material for subsequent rapid post-WGD selection.

Importantly, none of these tetraploid haplotypes were found in any diploid individuals, which indicates that they likely assembled upon the establishment of tetraploids. The rearrangement of diverse allelic sources into 'mosaic' tetraploid haplotypes may be facilitated by high recombination rates reported for these *Arabidopsis* species [56,57] and the observation of enhanced recombination rates in neo-tetraploid *A. arenosa* and *A. thaliana* [13,58]. Based on estimates of the age of the tetraploids of *A. arenosa* and *A. lyrata*, which range from approximately 20,000 to 230,000 generations [20,26,39], we speculate that the major tetraploid haplotype of each PSG rapidly spread across Europe. We provide hypotheses about the spatio-temporal context of the origin of these 'mosaic' haplotypes in S3 Text.

Altogether, we show that extensive reshuffling of trans-specific, species-specific, and novel variation occurred after WGD. This study demonstrates that polyploid lineages leverage their

enhanced capacity to accumulate genetic variation from various sources to generate new, potentially advantageous haplotypes. Importantly, this process is not unique to polyploids, as diploids have also been reported to accumulate multiple adaptive changes into finely tuned haplotypes [34,36,59]. Recent research in both plants and animals reveals substantial variation in the contributions of standing, introgressed, and de novo alleles to adaptation at the entire gene level [31–33,60]. The diversity of allelic sources in individual haplotypes in tetraploid *Arabidopsis* implies that pathways to adaptation may be even more diverse when considering the assembly of individual variants into haplotypes.

## Material and methods

### Sampling

*Arabidopsis*, a well-studied plant model genus, is primarily diploid, but autotetraploids have been discovered among the genetically diverse outcrossing species *A. arenosa* and *A. lyrata* in Central Europe. While *A. arenosa* diploids and tetraploids are widespread, *A. lyrata* primarily occupies a narrow ecological niche in Central Europe. Previous research suggested that introgression between *A. arenosa* and *A. lyrata* tetraploids facilitated the genetic differentiation of *A. lyrata* tetraploids from their diploid ancestors, allowing them to expand their niche [20,38].

Here, we sampled and sequenced genomes from both diploid and neighboring tetraploid populations across all known diploid-tetraploid lineages in Central European *Arabidopsis*. Our dataset includes genomes from total 73 newly sequenced diploid and tetraploid individuals of both species. Additionally, we incorporated data from 910 previously published whole genome sequences [20,26,31,32,50,53,61–65], encompassing a total of 983 individuals and 129 populations (S1 and S2 Data). The ploidy level of each individual in our new sampling was confirmed using flow cytometry following [66].

### Population genetic structure

Samples were whole genome resequenced (as detailed in S1 Data), and single nucleotide polymorphisms (SNPs) were called using a ploidy-aware approach following [26,31]. *Arabidopsis arenosa* and *A. lyrata* autotetraploids show random segregation of chromosomes with a tendency towards bivalent formation [5,20,35,39]. Bivalent formation during metaphase I of meiosis in established autotetraploids results from reduced crossover frequencies relative to neotetraploids or diploid relatives [3]. This results in allele frequency estimation comparable to diploids [67] and allowed to use a set of methods designed for diploids, which are also applicable to mixed-ploidy data; for a discussion see [68]. Specifically, we inferred relationships between populations using allele frequency covariance graphs implemented in TreeMix v.1.13 [40]. *Arabidopsis arenosa* was rooted with the diploid Pannonian population BDO and *A. lyrata* with the diploid Scandinavian population LOM. To obtain confidence in the reconstructed topology of *A. arenosa* and *A. lyrata* (Figs 1B and 1C), we performed a bootstrap analysis with 1000 bp blocks (matching the selection scan window size below) and 100 replicates. Further, we used Bayesian clustering in fastSTRUCTURE [42]. We randomly sampled two alleles per tetraploid individual using a custom script. This approach does not appear to bias clustering in autotetraploid samples based on [69]. Finally, we displayed genetic relatedness among individuals using principal component analysis (PCA) as implemented in the 'adegenet' package [70] in R. We calculated genome-wide four-fold degenerate (4d) within-population metrics (nucleotide diversity ($\pi$) and Tajima's D [71]) using the python3 'ScanTools_ProtEvol' pipeline [31].

To test for introgression, we used the D-statistics (ABBA-BABA statistics) as described in [43]. We assessed introgression between tetraploid *A. arenosa* (P2) and each *A. lyrata* tetraploid lineage (P3). For the non-admixed population (P1), we used the early diverging diploid

Pannonian lineage of *A. arenosa* [72], population BDO. Allele frequencies were polarized using diploid *A. halleri*, population GUN from Austria.

## Genome-wide scans for positive selection

To find candidate positively selected genes (PSGs) which likely contributed to adaptation to WGD in tetraploids, we employed two selection scan methods suitable for both diploids and autotetraploids following our best practices [68]. These methods are based on metrics estimated from population allele frequencies that are comparable across diploid and tetrasomic autotetraploid populations, in contrast to methods relying on assumptions on individual genotypes that are ploidy-specific [44,68].

First, between each of the tetraploid populations and the most proximal diploid population within each lineage, we calculated Fst [73] in 1 kb windows. We then used PicMin [44] to test whether there was evidence of repeated genetic differentiation among the tetraploid/diploid population pairs. PicMin uses order statistics to test whether population genetic summary statistics (in this case Fst) for orthologous genomic regions in different lineages exhibit a common shift towards extreme values in multiple lineages, indicative of repeated action of positive selection. PicMin was applied on windows that had data for at least three lineages (86,249 windows in total). A genome-wide false discovery rate correction was then performed with a significance threshold of q < 0.01. In cases of outlier signal spanning adjacent windows, the window with the lowest q-value and highest Fst was retained.

We also employed a 'candidate SNP' approach by calculating SNP-based Fst between each of the tetraploid populations and their most proximal diploid population within each lineage. A 1% outlier threshold was used to identify highly differentiated SNPs ('candidate SNPs' hereafter) across the genome. We determined the density of these candidate SNPs per gene using *A. lyrata* gene models [74]. Candidate genes were identified as the upper quartile with the highest density of outlier SNPs. Fisher's exact test ('SuperExactTest' package [75] in R) was then applied to verify repeatedly identified candidates. Notably, all PicMin-identified genes were confirmed by this candidate SNP approach.

## Functional annotation

To infer functions significantly enriched in our list of tetraploid PSGs, we performed gene ontology (GO) and UniProt Keywords enrichment analyses using the STRING database (last accessed 02/03/2023, [46]). We used *Arabidopsis thaliana* orthologs of *A. lyrata* genes. Only categories with FDR < 0.05 were considered. We also manually searched for functional descriptions of each gene using the TAIR database and associated literature (last accessed 02/03/2023, [47]). To identify protein-protein interactions, we used the STRING database, including all available information sources, and focused on 1st shell interactions.

## Haplotype block reconstruction

Due to unreliable standard phasing algorithms for short-read tetraploid samples [76], we first used a SNP frequency-based approach to reconstruct 'haplotype blocks' (here defined as linked, highly differentiated SNPs, which were later in the study confirmed with long-read data, further referred to as 'haplotypes'). We reconstructed the haplotypes for the 12 PSGs (a subset of 17 PSGs supported by at least five candidate highly differentiated SNPs) within *A. lyrata* and *A. arenosa* populations (232 candidate SNPs in total; S4 Data). We followed a procedure by [7]; briefly, haplotype block frequencies (HBFs) were calculated separately for diploids (218 *A. arenosa* and 121 *A. lyrata* individuals) and tetraploids (479 individuals). For each of the 12 PSGs with *n* candidate SNPs (for *n* see Fig 3B), we determined the allele frequency (*AF*)

of the major allele across all individuals. With 1916 tetraploid haplotypes and 436 (*A. arenosa*) / 242 (*A. lyrata*) diploid haplotypes in our dataset, the major *AF* corresponds to the tetraploid allele. We defined the tetraploid haplotype block frequency as the minimum among *n AF*s (HBFt = min {major *AF*}), and the frequency of diploid haplotype blocks as HBFd = 1 – max {major *AF*}. The frequency of all other haplotype blocks resulting from recombination or de novo mutations was defined as HBFr = 1 – HBFt – HBFd. Calculations were performed using an in-house R script.

We next validated our haplotype block analysis by estimating the correspondence between our allele frequency-based inferred haplotype blocks and (1) sequences from PacBio HiFi-based long read assemblies of the 12 PSGs in five diploid and five tetraploid samples of *A. arenosa* (see S9 Data for metadata), and (2) sequences from long reads themselves spanning two PSGs in two diploid and two tetraploid *A. arenosa* samples. For (1), we used BLASTn v.2.10.0 to extract sequences from the 12 PSGs in newly generated diploid and tetraploid assemblies. These sequences were then aligned using MUSCLE and visualized with IGV v.2.11.9 (S5 Data). For (2), we aligned the long reads spanning the region of the PSGs *CYCA2;3* and *PDS5b* to the assemblies, visualized with IGV, and manually exported the information about physical linkage of candidate SNPs into (S7 Data). As the PacBio HiFi reads from five diploid and five tetraploid individuals of *A. arenosa* confirmed the linkage across candidate SNPs, we call them haplotypes. This, however, does not mean that all tetraploids share the very same DNA molecule between the first and last SNP of a haplotype block, which would strictly mean haplotype, as there are lots of rare SNPs in between the linked candidate SNPs.

## Allelic sources of tetraploid haplotypes across the candidate positively selected genes

To assess if the tetraploid haplotypes originated from a single source or as a mosaic of allelic sources, we compiled a collection of genomes from 983 individuals: 818 individuals from 46 diploid and 61 tetraploid populations of *A. lyrata* and *A. arenosa*, and 165 individuals from four congeners. We employed two methods: tracing the evolutionary origins of candidate SNPs across the ancestral diploid lineages [50], and reconstructing networks of genetic distances at the PSG regions [70].

To determine the allelic sources contributing to tetraploid haplotypes, we investigated the presence/absence of the variants defining them (the 232 'candidate SNPs') within diploid individuals. We used a comprehensive genomic dataset encompassing diploid *A. lyrata* and *A. arenosa* (339 diploid individuals) alongside their outcrossing relatives *A. halleri*, *A. croatica*, *A. cebennensis*, and *A. pedemontana* ('diploid congeners' or 'outgroup', totaling 165 individuals; see S1 Data). A rarefaction analysis in [7] indicated that sampling as few as 40 diploid individuals across the *A. arenosa* species range captures the majority of diploid diversity. Therefore, our dataset of 504 diploid individuals should suffice to cover the full natural diversity of diploids. We excluded singletons (variants occurring only once in a species) to reduce the impact of sequencing errors (results were robust with or without singletons; S6 Fig). Then, using species' phylogenetic relationships ([51], Fig 4A), we determined likely allelic sources for the 232 candidate SNPs among 504 diploid individuals. We categorized each SNP according to one of the following seven scenarios (Fig 4A): 1) trans-specific polymorphism in both *A. lyrata* and *A. arenosa*; 2) trans-specific polymorphism in *A. arenosa* only (tetraploid SNP occurs in *A. arenosa* but not *A. lyrata* diploids and in one to all diploid congeners); 3) trans-specific polymorphism in *A. lyrata* only (tetraploid SNP occurs in *A. lyrata* but not *A. arenosa* diploids and in one to all diploid congeners); 4) trans-specific polymorphism in the congeners only; 5) *A. arenosa*-specific polymorphism; 6) *A. lyrata*-specific polymorphism; 7) tetraploid-private

polymorphism. Here, we differentiate between trans-specific and species-specific variability at the diploid level, reflecting ancestral allele sharing rather than recent introgression. It is crucial to note that any tetraploid SNP identified here ultimately represents trans-specific polymorphism since it is shared between tetraploids of both *A. arenosa* and *A. lyrata*.

As additional evidence to estimate variability in allelic sources of tetraploid haplotypes, we constructed genetic distance networks using Nei's distance [77] in the 'adegenet' package in R [70]. We visualized these networks using SplitsTree [41].

## Allelic sources of the genomic background

To test if the allelic sources of the 232 candidate SNPs for adaptation to WGD differed from the genomic background, we estimated the relative contribution of the same allelic sources, in particular the seven scenarios outlined above, to synonymous variation from the genome-wide background. The following steps were undertaken to make this analysis comparable with the analysis of the 232 candidate SNPs: All variable sites within the core dataset of 155 Central European *A. arenosa* and *A. lyrata* (depth $\geq$ 4, missing data $\leq$ 60%) were extracted. From these total 5,092,575 SNPs, a BED file of synonymous SNPs was generated, resulting in total 1,341,033 SNPs. Then, SNPs were extracted from this BED file, applying a filter for missing data $\leq$ 20%, i) from a dataset comprising the 871 individuals of *A. arenosa* and *A. lyrata* only (1,198,065 SNPs), and ii) from a dataset of 173 individuals from the outgroup (339,761 SNPs). The obtained SNPs from both datasets were overlaid, sites which were invariant across all tetraploids removed, and a random subset of 232 SNPs across the genome was drawn 1000 times. To prevent overestimation of the proportion of trans-specific polymorphism, SNPs missing in the outgroup dataset were treated as invariant. The SNP data subsets were polarized towards tetraploids of *A. arenosa* and *A. lyrata*. Allelic source scenarios for each of the 232 SNPs were inferred in the same way as for the 232 candidate SNPs for adaptation to WGD.

## Supporting information

**S1 Fig. Population structure of *A. arenosa* (A) and *A. lyrata* (A-D).** A: PCA shows separation between *A. arenosa* and *A. lyrata* over 50,000 of scaffold_1 SNPs. (B-D): PCA, Nei's distance-based neighbor-joining tree, and fastSTRUCTURE plot all support three lineages of tetraploid *A. lyrata*, named AL Germany, AL Czechia, and AL Austria. For these analyses we used 1,094,553 genome-wide four-fold degenerate SNPs. Red: diploid populations, blue: tetraploid populations.
(EPS)

**S2 Fig. Location of candidate genes (black vertical lines) on *A. lyrata* reference chromosomes colored by bins of distinct recombination rate per gene.** The figure indicates that gene candidates do not cluster in regions with extreme values of recombination rate per gene (blue), as estimated based on the available *A. lyrata* recombination map [56].
(EPS)

**S3 Fig. Candidate positively selected genes (PSGs) involved in processes of cell cycle regulation, meiosis, and transcriptional regulation vary in their proportion of differentiated cis-regulatory, nonsynonymous, and synonymous SNPs.** The number above each bar corresponds to the number of differentiated SNPs within each gene. ~5M of genic and cis-regulatory SNPs from whole genome sequencing data were taken as controls.
(EPS)

**S4 Fig. Phenotypic shift associated with the establishment of tetraploids of *A. arenosa* and *A. lyrata* in the form of a decrease in the level of DNA endoreduplication.** Top: Proportion

of endoreduplicated nuclei in leaves of 10 individuals per each lineage and ploidy. **: p < 0.01, ****: p < 0.0001, ns: nonsignificant, Wilcoxon rank sum test. The horizontal violet line shows the level of endoreduplication under a scenario of no post-WGD adaptation, estimated from values of synthetic neo-tetraploids of *A. arenosa*. Bottom: Prediction for the compensatory evolution of polyploid traits from [2] and observed average DNA content per leaf cell (calculated as a mean number of homologous chromosomes per nucleus) for diploid, neo-tetraploid, and tetraploid *A. arenosa*.
(EPS)

**S5 Fig. Comparison of leaf endoreduplication among diploids.** Upper boxplots show the endoreduplication level, calculated as the number of endoreduplicated nuclei divided by all nuclei in the analysis, bottom plots show the maximum number of endoreduplication cycles in the leaf per each diploid lineage. ****: p < 0.0001, ***: p < 0.001, ns: nonsignificant, Wilcoxon rank sum test. Each boxplot is represented by 10 individuals.
(EPS)

**S6 Fig. Variable allelic sources of tetraploid haplotypes for each of the 12 tetraploid positively selected genes (PSGs).** Barplots show the proportion of candidate SNPs representing each of the seven source scenarios (see Fig 4A for graphical visualization of scenarios). Upper plot shows the source when including singletons, bottom plot after filtering singletons out.
(EPS)

**S7 Fig. Nei's distance-based neighbor-joining networks of tetraploid positively selected genes (PSGs) across diploid *Arabidopsis* (*A. arenosa*, *A. lyrata*, *A. croatica*, *A. cebennensis*, and *A. pedemontana*) and tetraploid *Arabidopsis* (*A. arenosa*, *A. lyrata*).** Tetraploids from all four tetraploid lineages of *A. arenosa* and *A. lyrata* (blue) always form a single lineage, suggesting the presence of a single shared tetraploid haplotype at the locus. Further, they form an unresolved network with diploids of multiple species (red), suggesting diversity of allelic sources of each shared tetraploid haplotype ('mosaic scenario').
(EPS)

**S1 Table. Genome-wide nucleotide diversity and Tajima's D of *A. lyrata* populations newly sequenced here, calculated over four-fold degenerate sites.**
(DOCX)

**S2 Table. Summary of the 17 candidate tetraploid positively selected genes (PSGs).**
(DOCX)

**S3 Table. Functional annotation of the 17 tetraploid positively selected genes (PSGs).**
(DOCX)

**S4 Table. Presence and frequency of tetraploid, two diploid, and other haplotype blocks in all 61 tetraploid populations of *A. arenosa* and *A. lyrata* (479 individuals).** AF: allele frequency of haplotype.
(DOCX)

**S5 Table. Variable allelic sources of tetraploid haplotypes for 1000-times randomly sampled 232 synonymous SNPs representing the genomic background versus the 232 candidate SNPs of the 12 positively selected genes (PSGs).** The proportion of SNPs representing each of the seven source scenarios is given (see Fig 4A for graphical visualization of scenarios).
(DOCX)

**S1 Data. Metadata and sequence quality checks for the 983 whole genome sequenced individuals.**
(XLSX)

**S2 Data. Sampling locations of the 129 populations.**
(XLSX)

**S3 Data. Set of 54 significant PicMin windows and genome-wide PicMin results underlying the PicMin Manhattan plot.**
(XLSX)

**S4 Data. List of outlier genes identified using the 'candidate SNP' approach and their overlap among tetraploid lineages.** Candidate SNPs were used in the selection scan to identify positively selected genes (PSGs), following a three-step procedure. First, 1% outlier SNPs were identified; second, the top quartile of genes with the highest density of outlier SNPs were identified; and third, these genes were overlapped among the four tetraploid lineages of *A. arenosa* and *A. lyrata* to identify repeatedly differentiated genes. The PSGs and their overlap are shown here.
(XLSX)

**S5 Data. Sequences of the 12 positively selected genes (PSGs), assembled using long read sequencing of five diploid and five tetraploid individuals.**
(XLSX)

**S6 Data. Table of linked candidate SNPs, as determined using long read sequencing.** Position of variants within the same column (columns *D-N*) highlights their physical linkage within a gene region (black boxes). Note that these linked candidate SNPs were used as markers to reconstruct haplotypes from short read data, which is summarized in columns *Q-AMJ*.
(XLSX)

**S7 Data. Candidate SNPs marking haplotypes of *CYCA2;3* and *PDS5b* (Figs 4C and 4D), as found on the same long read.**
(XLSX)

**S8 Data. Distribution of the 232 tetraploid candidate SNPs among the 504 diploid samples used to estimate the likely sources of tetraploid haplotypes.**
(XLSX)

**S9 Data. Metadata and sequence quality checks for the 10 long read-sequenced individuals using PacBio HiFi.**
(XLSX)

**S1 Text. Detailed functional interpretations of the candidate positively selected genes (PSGs).**
(DOCX)

**S2 Text. Detailed methods and results of estimating DNA endoreduplication using flow cytometry.**
(DOCX)

**S3 Text. Hypotheses about the spatio-temporal context of the origin of the 'mosaic' haplotypes.**
(DOCX)

## Acknowledgments

We greatly appreciate the constructive feedback from members of the Ecolgen team in Prague, Katie Peichel, Reto Burri, Alison Scott, Polina Novikova, Andrew MacColl, Tuomas Hämälä, John Brookfield, Emma Curran, Laura Dean, Sian Bray, and Ana da Silvia. We further appreciate inspiration provided by the ForBio Polyploid course in Drøbak, Norway. We thank Martin Čertner and Dorka Čertnerová for sharing their polyploid synthesis protocol, Bodo Schwarzberg (Untere Naturschutzbehörde Nordhausen) for sharing seeds of the STD population, and Bertram Preuschhof (Untere Naturschutzbehörde Göttingen) for a collecting permit for the SCT population. Sequencing was performed by the Norwegian Sequencing Centre, University of Oslo. Computational resources were provided by the CESNET LM2015042 and the CERIT Scientific Cloud LM2015085, under the program Projects of Large Research, Development, and Innovations Infrastructures.

## Author Contributions

**Conceptualization:** Magdalena Bohutínská, Levi Yant, Filip Kolář, Roswitha Schmickl.

**Data curation:** Magdalena Bohutínská, Jakub Vlček.

**Formal analysis:** Magdalena Bohutínská, Eliška Petříková, Cristina Vives Cobo, Jakub Vlček.

**Funding acquisition:** Karol Marhold, Levi Yant, Filip Kolář, Roswitha Schmickl.

**Investigation:** Magdalena Bohutínská, Eliška Petříková, Cristina Vives Cobo, Gabriela Šrámková, Alžběta Poupětová, Jakub Hojka.

**Methodology:** Magdalena Bohutínská, Eliška Petříková, Tom R. Booker.

**Project administration:** Roswitha Schmickl.

**Resources:** Filip Kolář, Roswitha Schmickl.

**Supervision:** Levi Yant, Filip Kolář, Roswitha Schmickl.

**Visualization:** Magdalena Bohutínská.

**Writing – original draft:** Magdalena Bohutínská, Levi Yant, Filip Kolář, Roswitha Schmickl.

**Writing – review & editing:** Eliška Petříková, Tom R. Booker, Cristina Vives Cobo, Jakub Vlček, Gabriela Šrámková, Alžběta Poupětová, Jakub Hojka, Karol Marhold.

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
