## [Decision Letter · Decision Letter 0]

12 Sep 2024

Dear Dr Schmickl,

Thank you very much for submitting your Research Article entitled 'Polyploids broadly generate novel haplotypes from trans-specific variation in Arabidopsis arenosa and Arabidopsis lyrata' to PLOS Genetics.

The manuscript was fully evaluated at the editorial level and by independent peer reviewers. The reviewers appreciated the attention to an important topic but identified some concerns that we ask you address in a revised manuscript.

We therefore ask you to modify the manuscript according to the review recommendations. Your revisions should address the specific points made by each reviewer.

To resubmit, log into your Editorial Manager account and select the option 'Revise Submission' in the 'Submissions Needing Revision' folder.

Yours sincerely,

Yalong Guo, Ph.D.

Guest Editor

PLOS Genetics

Justin Fay

Section Editor

PLOS Genetics

The in-depth review of your manuscript by the three reviewers is now complete. Based on their assessment, it is clear that your manuscript requires some revision before it can be considered further for publication in PLOS Genetics. Comments from the external reviewers are included below.

In particular, especially please pay attention to the following issues:

1) As reviewer1 pointed out "The aspect that may seem a bit surprising to some readers and that the authors do not stress is the fact that most genome scans studies comparing diploids and tetraploids end up finding roughly the same set of genes. Those are few and related to the same general function." It is important to discuss this "Are those the only changes or are we missing the rest?".

2) As reviewer2 pointed out that the importance of endopolyploidy in this study should be rephrased somehow.

Reviewer's Responses to Questions

**Comments to the Authors:**

Reviewer #1: This is an interesting and carefully carried out study. The paper is well written but, perhaps misses what seems, to this reader at least, one of the main conclusions/caveats. Many of the authors of the present manuscript have been working for a rather long time on tetraploid Arabidopsis species and have made important contributions, in particular to our understanding of the importance of genes related to meiosis during the shift from diploidy to tetraploidy. This paper is an additional contribution to this line of work. In previous studies, A. arenosa and A. lyrata were studied separately. Building on a massive dataset the authors study them jointly and highlight the importance of introgression, from other species, in the evolution of genes differentiating the two ploidy levels. Their work confirms that polyploids can act as "bridges" where hybridization between diploids is impossible. As they point out in the discussion, this is not an isolate case and there have recently been other studies pointing out the importance of introgression to generate potentially beneficial diversity within polyploid species complexes. The aspect that may seem a bit surprising to some readers and that the authors do not stress is the fact that most genome scans studies comparing diploids and tetraploids end up finding roughly the same set of genes. Those are few and related to the same general function. Now, not so recently the prevailing views in the literature on the effect of Whole genome doubling were more in line with a "shock and awe" event than subtle and limited changes. Many recent studies have been more in line with the latter than with the former. It could be that this is simply a consequence of the methods used to compare diploids and polyploids, but in any case, I feel it would be worth pointing this out in the discussion. Are those the only changes or are we missing the rest? Although I would tend to think the latter holds, in my understanding the jury is still out.

Minor comments:

1. line 71: removal or weakening of hybridization barriers?

2. line 74. In the pre-genomic era Slotte et al. MBE 2008 was a rather convincing example of introgression from diploids to tetraploids rather than the converse.

3. line 159-163. Perhaps a bit more information could be lifted from the supplementary file.

Reviewer #2: This manuscript presents a comprehensive study of genetic variation contributing to repeated adaptation in autopolyploid Arabidopsis lysate and A. Arenosa relative to their diploids. The authors employed genome-wide selection scanning methods to detect the genes and processes under selection across four tetraploid lineages, and next focused on 17 PSGs to characterize types of variants. Regarding haplotypes, the authors found most tetraploid haplotypes are novel from those in diploids. Regarding SNPs, the extent of different evolutionary sources were measured: trans-specific polymorphism from diploids (65.5%), new mutations (31.9%), tetraploid-specific inter-species hybridization (71.1%). These diverse SNP sources contribute to the novel, mosaic haplotypes bearing the repeated selection signatures of tetraploid adaptation.

This is a very well written manuscript and I appreciate the concise presentation of key findings. The authors also provide thorough supplemental materials to cover technical details and additional exploration such as the functional interpretation of candidate PSGs. I would like to offer only a few suggestion for improvement:

* Methodological Choice: While the authors used two genome-wide selection scanning methods, it would be beneficial to discuss the rationale for choosing these methods over other popular options like McDonald-Kreitman tests. Additionally, please elaborate on the suitability of these methods for diploid-autopolyploid species.

* Please include data information of PacBio HiFI reads used.

* L183: Clarify whether the term "haplotype" refers to haplotype blocks or physical haplotypes confirmed by HiFi reads.

* The higher proportion of trans-specific polymorphism sourced from A. arenosa compared to A. lyrata is intriguing. Besides the basal placement and larger genetic pool of A. arenosa, are there other possible explanations? For example, direction biased gene flow or selection for functional alleles from A. arenosa, etc.

* L448: Fig 5A should be Figure 4A

* L228-236: For the genomic background analysis, provide statistical test significance results from the permutation test to support the claim of lower trans-specific polymorphism, introgression, and new mutations.

* Only 11 PSG sequences were provided in TableS5.

Reviewer #3: This is a compact, well-written manuscript presenting some very interesting results. For me it was a good wake-up call to be more attentive to the Arabidopsis polyploidy system than I have been. Despite knowing some of the story from these authors concerning evolution involving meiosis-related genes in polyploids, it is clear that I am woefully behind in reading about the overall Arabidopsis evolutionary picture, particularly the issue of trans-specific variation involving wide hybridization. That, of course, plays a key role in the research reported here. The population genomic resources used in the study are quite impressive—the envy of those not working on near-model species—and are leveraged to excellent effect in this study.

The authors report that autopolyploids classified as A. lyrata or A. arenosa, though bearing different names, share haplotypes through post-polyploidy hybridization/introgression. That is a very interesting finding, one that I was more familiar with from other systems and from theoretical considerations, nicely demonstrated here.

Next, they identify 17 genes that are under positive selection in the polyploids (PSGs); these are either related to meiosis or to other cellular-level phenomena (notably cell cycle). This makes sense given previous work. What I found (find) difficult to understand fully was the emphasis, particularly in the Supplementary text, on endopolyploidy. I do not know the literature on the significance of endopolyploidy in natural Arabidopsis populations that is cited here. I do know something about endopolyploidy in synthetic A. thaliana polyploids (e.g., Corneille et al. 2017, Plant Physiology; Robinson et al. 2018, Plant Cell), but here the argument seems to be that the degree of endopolyploidy is associated with adaptation. This should be explained better (see comment below), but I seem to have gotten the main interesting points of the study without understanding this fully—there are 17 genes that show signs of positive selection in the polyploids in a way that is consistent with adaptation to an altered cellular landscape in which function is challenged by the presence of a doubled genome.

The authors then show that for the 12 genes with sufficient SNPs, tetraploids possess a major haplotype not found in diploids, including putative diploid source populations. Strikingly, these tetraploid-specific haplotypes have sequences that are a mosaic of different sources—not only the two diploid cytotypes of the same two species whose names the polyploids bear or from de novo mutations in the tetraploids, but in a significant percentage of cases coming from other Arabidopsis species. SNPs in each gene are traced to their sources in one of 7 scenarios. Fig. 4C is very helpful in illustrating what was done to produce the data summarized in 4B; 4A took a bit of time to decipher, but I don’t have any good suggestion of how to make it more clear (I think part of the problem may be my interpretation of “standing in diploids”, which can mean as few as a single species, and might be clearer with a longer description such as “standing in one or more diploid species” … but there isn’t much room in this already crowded, compact figure). Networks for each gene are presented in Supplementary Fig. 7, which I found very useful (see comment below).

One thing that seemed lacking here was an indication of where the SNPs under selection, identified in the “candidate SNP” approach that identified all of the PICMIN genes plus three additional candidates. Presumably at least some of these are among those shown for the two genes in Fig. 4C, and all of them can be placed in one of the 7 scenarios. I thought I might find that information in Supplemental Dataset 3, but (as noted below) I found that spreadsheet undecipherable.

In the Discussion, the authors present a narrative involving origins of the two polyploids and subsequent introgression, with selection on genes contributing to meiotic (or more broadly, cytological) stability. I have already commented on my inability to understand how this relates to endopolyploidy, which is a perfectly natural part of the normal functioning of some plant cell types. It seems to me that the endopolyploidy part is—or at least comes across as—a rather peripheral aspect of this paper, disconnected from the major points that are made.

In lines 311-312 the authors state, “Altogether, we demonstrated that extensive reshuffling of trans-specific, species-specific, and novel variation occurred in response to the challenges of WGD.” Most of this seems well-supported by the results, but “in response to” sounds rather teleological, as though the polyploid was looking for a way to adapt. Perhaps I am being too critical, but I wondered at one point whether genes other than the 12 PSGs studied here are also mosaics. If so, then the combining of various sources of variation is not “in response to” anything in particular—it simply is a product of the biology of these Arabidopsis lineages. The authors may be admitting the same thing by saying that this kind of mosaicism is not confined to polyploids, but also in diploids (lines 315-316) where clearly it is not “in response to the challenges of WGD”, at least not in the immediate past. Given the variation in the distribution of sources of SNPs across the 12 genes (Fig. 4B), and the similarity of the “genome” distribution in the same figure, I would be surprised if non-PSGs didn’t show the same range of variation in terms of sources of SNPs.

None of this diminishes the value of this work, which I think is of considerable interest, and as best as I can tell has been executed competently. I think it will be a useful contribution to the area of polyploid evolution.

Specific comments, by line number:

Line 59. “Autopolyploid” is here defined in the commonly used taxonomic sense, which I would argue is genetically misleading. There is an excellent very recent paper on this topic from Anneliese Mason (one of the stars in the polyploidy field) that I think the authors may want to cite here:

Lv, Z., Addo Nyarko, C., Ramtekey, V., Behn, H., and Mason, A.S. (2024). Defining autopolyploidy: Cytology, genetics, and taxonomy. American Journal of Botany 111, e16292.

117ff. I’m a bit puzzled by the lack of any evidence of reticulation in the TreeMix figures (2B, 2C); this isn’t a program I have used recently, but my recollection is that it is used to show gene flow (e.g., via introgression), which is clearly shown later in Fig. 2D. The legend has no explanation of how to interpret the topologies of these figures, or perhaps the lack of cycling in the graph.

148. Dataset 3. I looked at this spreadsheet, and could not make much sense of it. It would be very helpful if an explanation of some kind was included somewhere, preferable at the top of the file as in a regular text table.

159-160. “PSGs associated with the cell cycle trait endoreduplication on tetraploids”: This is awkwardly phrased. Maybe add commas or quotes?

173-174. “tetraploids aggregate as mosaics from different allelic sources (Fig. 1B)”:

250. “polyploidy-mediated hybrid seed rescue” is not a term with which I am familiar, at least by name. I took a quick look at Ref. 15 to see what is meant, and found that it relates to endosperm balance number. If there is a way to explain the phenomenon succinctly here instead of using what is a rather jargon-ish term, that could be helpful to readers like me.

267-268. “We found a substantial decrease in endoreduplication within established tetraploids compared to newly synthesized tetraploids.”: What is the adaptive significance of reduced endoreduplication? It is a normal part of development in many species, in which case it presumably is beneficial. My sense from the literature is that the effect of whole-plant doubling on endopolyploidy is variable.

299. “retainment”: Probably "retention" would be a better word here.

343-344. “Arabidopsis arenosa and A. lyrata autotetraploids show random segregation of

chromosomes [32, 38], resulting in allele frequency estimation comparable to diploids”: Random segregation in an autopolyploid should produce tetrasomic ratios, not disomic ratios as in diploids. Consequently I’m not sure what is meant here. See Lv et al. (2024) for discussion of the relationships among segregation ratios, bivalents/multivalents, and auto/allopolyploidy.

Supplementary text 1: “In summary, these results point to an adaptive compensation for DNA content per nucleus: while nearly double in neo-tetraploids, it returns toward diploid levels in established tetraploids (Fig. S4).” Genome downsizing references (e.g., Leitch and others) could be useful here; this is not a new concept in the polyploidy literature.

**Have all data underlying the figures and results presented in the manuscript been provided?**

Reviewer #1: Yes

Reviewer #2: Yes

Reviewer #3: Yes

PLOS authors have the option to publish the peer review history of their article (what does this mean?). If published, this will include your full peer review and any attached files.

Reviewer #1: No

Reviewer #2: No

Reviewer #3: **Yes: **Jeff J. Doyle

---

## [Decision Letter · Decision Letter 1]

28 Nov 2024

Dear Dr Schmickl,

We are pleased to inform you that your manuscript entitled "Polyploids broadly generate novel haplotypes from trans-specific variation in Arabidopsis arenosa and Arabidopsis lyrata" has been editorially accepted for publication in PLOS Genetics. Congratulations!

Yours sincerely,

Yalong Guo, Ph.D.

Guest Editor

PLOS Genetics

Justin Fay

Section Editor

PLOS Genetics

Aimée Dudley

Editor-in-Chief

PLOS Genetics

Anne Goriely

Editor-in-Chief

PLOS Genetics

Comments from the reviewers (if applicable):

Reviewer's Responses to Questions

**Comments to the Authors:**

Reviewer #1: I have read the replies to the comments of the three reviewers as well as the revised manuscript and I feel that the authors have taken into account all of our comments and that the manuscript can now be published.

Reviewer #2: The authors have satisfactorily addressed my concerns.

Reviewer #3: The authors have addressed the concerns I raised in my review of the initial submission. The only substantive issue concerned endopolyploidy, and I feel that their response to my review was acceptable.

The study as described in the initial submission was well-conceived and executed, with convincing results and appropriate discussion. I believe it to be a very useful contribution to the literature on polyploid evolution.

**Have all data underlying the figures and results presented in the manuscript been provided?**

Reviewer #1: Yes

Reviewer #2: Yes

Reviewer #3: Yes

PLOS authors have the option to publish the peer review history of their article (what does this mean?). If published, this will include your full peer review and any attached files.

Reviewer #1: No

Reviewer #2: No

Reviewer #3: **Yes: **Jeff J. Doyle

**Data Deposition**

http://datadryad.org/submit?journalID=pgenetics&manu=PGENETICS-D-24-00791R1

**Press Queries**

---

## [Editor Report · Acceptance letter]

16 Dec 2024

PGENETICS-D-24-00791R1 

Polyploids broadly generate novel haplotypes from trans-specific variation in Arabidopsis arenosa and Arabidopsis lyrata 

Dear Dr Schmickl, 

We are pleased to inform you that your manuscript entitled "Polyploids broadly generate novel haplotypes from trans-specific variation in Arabidopsis arenosa and Arabidopsis lyrata" has been formally accepted for publication in PLOS Genetics! Your manuscript is now with our production department and you will be notified of the publication date in due course.

With kind regards,

Dorothy Lannert

PLOS Genetics

On behalf of:
